# Small Bowel Obstruction Masking a Perforated Dermoid Ovarian Cyst

**DOI:** 10.3390/diagnostics14171975

**Published:** 2024-09-06

**Authors:** Ismini Kountouri, Christos Gkogkos, Ioannis Katsarelas, Periklis Dimasis, Amyntas Giotas, Eftychia Kokkali, Miltiadis Chandolias, Nikolaos Gkiatas, Dimitra Manolakaki

**Affiliations:** 1Department of General Surgery, General Hospital of Katerini, 60132 Pieria, Greece; giannis24katsarelas@gmail.com (I.K.); dimasis@yahoo.com (P.D.); miltoshandolias@gmail.com (M.C.); nikgiat71@gmail.com (N.G.); dimanolakaki@gmail.com (D.M.); 2Gynecology and Obstetrics Department, General Hospital of Katerini, 60132 Pieria, Greece; akisgogos@yahoo.gr (C.G.); ammag10@live.com (A.G.); 3Department of Radiology, General Hospital of Katerini, 60132 Pieria, Greece; kokkalieutuxia@gmail.com

**Keywords:** small bowel obstruction, perforated ovarian cyst, generalized peritonitis

## Abstract

A 58-year-old female presented with abdominal pain, vomiting and constipation. Laboratory tests indicated elevated white blood cell count and C-reactive protein levels. Imaging via CT scan revealed a large cystic mass in the right ovary, abscesses and generalized small bowel distension, which initially raised suspicion of the existence of ovarian cancer with peritoneal carcinomatosis. Despite conservative management, the patient’s condition did not improve, prompting a laparotomy. Intraoperative findings included generalized peritonitis, significant small bowel dilation due to inflammatory adhesions and a perforated dermoid ovarian cyst. The cyst was resected and a prophylactic ileostomy was installed. Histopathological examination confirmed the diagnosis of a benign dermoid ovarian cyst. This case illustrates the rare presentation of a perforated dermoid cyst mimicking peritoneal carcinomatosis and emphasizes the importance of considering such complications in the differential diagnosis of bowel obstruction and peritoneal disease. Early recognition and appropriate surgical intervention are crucial for optimal outcomes.

A 58-year-old female patient presented to the Emergency Department of the General Hospital of Katerini in Pieria, Greece, with complaints of abdominal pain, vomiting and constipation. She reported no history of previous abdominal surgery, had no significant medical history and was not taking any specific medication. Laboratory tests indicated an elevated white blood cell count of 18.24 × 10^3^/μL and C-reactive protein values of 29.83 mg/dL. Physical examination revealed abdominal distension, guarding and rebound tenderness.

The patient was admitted to the Surgical Ward and underwent a Computed Tomography (CT) scan, with oral Gastrografin contrast, which revealed two abscess cavities—one below the right hemidiaphragm and one in the left paracolic gutter—as well as generalized distension of her small bowel, a large cystic mass in the right ovary and multiple peritoneal implants (Figure 1a–c). Due to the presence of peritoneal implants, a preliminary diagnosis of ovarian cancer with generalized peritoneal carcinomatosis was made. Initial treatment was conservative, consisting of intravenous fluids, a nasogastric tube and antibiotics. Two days later, with no improvement in the patient’s clinical condition, a second abdominal CT scan was performed, again demonstrating distended small bowel loops with no distal contrast passage. Following discussions with the patient and her family, and consultations among the medical, surgical and gynecological teams, the decision was made to proceed with a laparotomy the following day.

During the laparotomy, generalized peritonitis was observed, characterized by a large amount of fibrinous exudate and free purulent fluid. The small bowel was significantly dilated due to obstruction caused by multiple inflammatory adhesions. The adhesions were lysed and the peritoneal cavity was thoroughly explored. No signs of peritoneal carcinomatosis or point of small bowel obstruction were found. However, a cystic formation protruding from the right ovary was identified, with a rupture in its wall through which purulent fluid was leaking (Figure 2).

Two abscess cavities were identified: one located below the right hemidiaphragm and another in the left paracolic gutter (Figure 3a,b). A lavage of the peritoneal cavity was performed, along with an en bloc salpingo-oophorectomy and resection of the cystic formation. The cystic formation was submitted for histopathological examination. Due to the excessive small bowel dilation and the presence of small bowel ileus, a prophylactic loop ileostomy was installed in the right abdominal wall.

The patient had an uncomplicated postoperative course and awaited her histopathology results, which confirmed the presence of a dermoid ovarian cyst. No evidence of malignancy was found.

A dermoid ovarian cyst is the most common form of benign ovarian tumor [1,2,3]. These tumors account for 10–20% of all ovarian tumors [3] and are often asymptomatic and typically cause symptoms only when enlarged or if they become complicated [1]. Complicated ovarian cysts can be severe life threatening events, with ovarian torsion (16%), presenting mainly with pain, being the most common. Relatively less frequent complications include cystic rupture (1–2%), infection (1%) or malignant transformation (2%) [1,2]. A bowel involvement is particularly rare, with only a few cases documented in the literature [4]. Dermoid ovarian tumors may lead to bowel obstruction or the formation of an entero-ovarian fistula, both of which are infrequently reported in the literature [1,4], presenting with pain, rectal bleeding or the passage of dermoid cyst contents within the stool [3]. A perforated ovarian cyst can cause acute peritonitis subsequent to obstructive ileus of the small bowel [2].

Li et al. reviewed the literature and found that the size of a dermoid ovarian cyst can contribute to cystic rupture, with most ruptured cysts, including the case presented here, being 6 to 10 cm in diameter [5].

CT and Magnetic Resonance Imaging (MRI) are particularly sensitive to the presence of fat, and so the diagnosis of a dermoid ovarian cyst can be easily made in 93% of mature dermoid ovarian cysts cases that contain fat [2,4]. On CT, the tumor will demonstrate a component of fat density, usually alongside soft tissue and calcium components [4]. On MRI, the fat-containing component of the tumor will demonstrate a T1 hyperintense signal that suppresses T1-weighted chemical selective fat-saturated images [4]. CT findings also include evidence of rupture of sebaceous material into the peritoneal cavity, including fat–fluid layering ascites and fatty implants likely related to the chemical or granulomatous peritonitis induced by chronic leakage of the sebaceous material of the cyst [2]. Ultrasound imaging can be used for the diagnosis of dermoid ovarian cysts through identifying the presence of a dermoid plug, or Rokitansky nodule (a peripheral solid echogenic nodule with posterior acoustic shadowing projecting into the cystic cavity), an echogenic or partly echogenic mass due to its sebum and hair content, the presence of a dermoid mesh (innumerable thin echogenic lines suspended throughout the cyst cavity) and fluid–fluid levels formed by less dense echogenic sebum floating on more dense hypoechoic or anechoic aqueous material [4].

This case report highlights a rare instance of a perforated ovarian cyst with radiological findings that mimic generalized carcinomatosis. A perforated dermoid ovarian cyst can cause peritonitis [6], which may present either acutely, with symptoms of acute abdomen and shock [6], or, as in this case, chronically [7,8]. Chronic peritonitis due to a perforated dermoid ovarian cyst is more challenging to diagnose due to its subtle and non-specific symptoms [6]. CT scans are useful in diagnosis, revealing characteristic hypoattenuating fat–fluid levels in dependent pockets within the peritoneal cavity, typically below the right hemidiaphragm, which is a pathognomonic finding [6]. In chronic granulomatous peritonitis, multiple small white peritoneal lesions, dense adhesions, and ascites may mimic carcinomatosis or tuberculous peritonitis, and so these pathologies should be considered in the differential diagnosis [6]. Mild chronic abdominal pain, a very thin cyst wall remnant, fat–fluid levels, and foamy fat signals on MRI are important clues for the differential diagnosis of dermoid cyst rupture and chronic spillage [6]. An MRI scan can be helpful for the differential diagnosis [8]. This case report showcases that, sometimes, the radiological and clinical findings of a perforated ovarian cyst with chronic granulomatous peritonitis can lead to the misdiagnosis of generalized carcinomatosis and eventually delay the definitive treatment.

Dermoid ovarian cysts identified before rupture are usually managed surgically, either laparoscopically or via laparotomy [9]. The recurrence rate of mature cystic teratomas is less than 3–4%, with malignancy occurring in approximately 1.7% of cases [9]. Recent studies indicate that laparoscopic removal of ovarian dermoid tumors is a safe approach [10,11,12].

For ruptured dermoid ovarian cysts, surgical intervention is the treatment of choice [11,12]. Laparoscopic management can be effective for the treatment of ruptured dermoid ovarian cysts but is best performed by experienced surgeons [11]. Laparotomy remains the preferred treatment of choice in most cases [5,11,12]. In 2023, Takeda et al. reported that laparoscopic management was feasible in eight out of nine cases of ruptured dermoid ovarian cysts, except for one case with severe adhesion requiring laparotomy [12]. For generalized peritonitis, surgical treatment involves unilateral salpingo-oophorectomy and abdominal lavage with complete pelvic and abdominal cavity exploration, peritoneal washing and/or sampling of ascites [8,13]. Early surgical intervention can reduce complications, prevent new adhesions and nodules and facilitate a faster recovery [8].

This case report aims to increase awareness among surgeons and gynecologists regarding the rare occurrence of bowel obstruction due to complicated ovarian cysts. The fluid from a ruptured dermoid ovarian cyst can cause purulent peritonitis and peritoneal thickening, potentially leading to misdiagnosis as peritoneal carcinomatosis. It is crucial for healthcare providers to consider this rare cause of bowel obstruction in the differential diagnosis of patients presenting with relevant clinical and radiological findings.

## Figures and Tables

**Figure 1 diagnostics-14-01975-f001:**
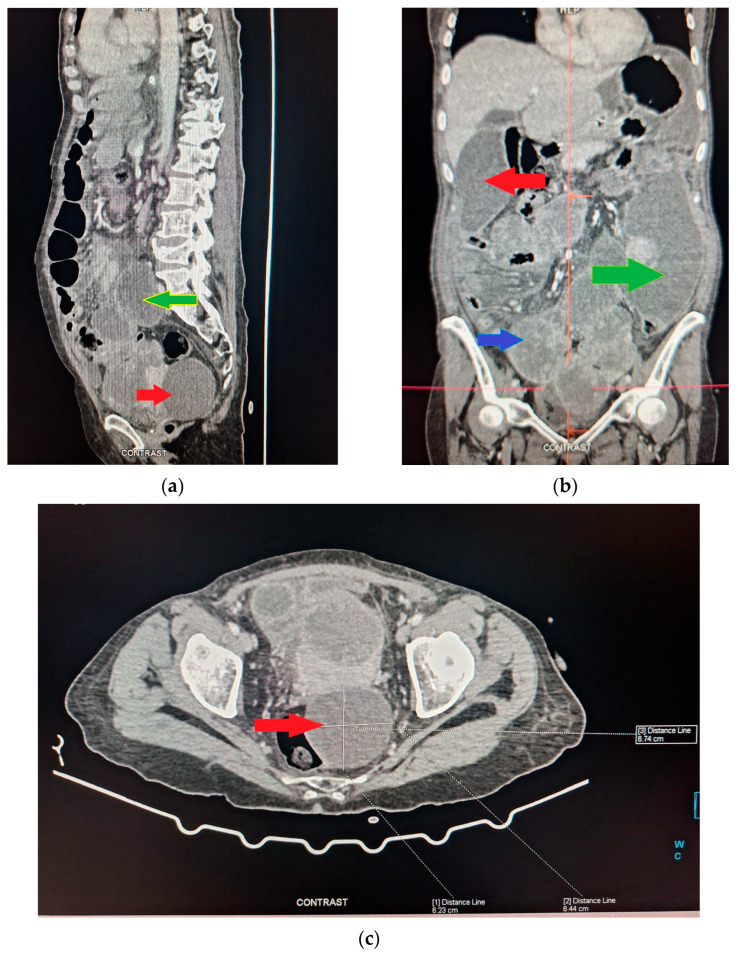
A contrast-enhanced abdominal CT scan of the patient. (**a**) The cystic mass is indicated with the red arrow and the small bowel dilation with the green arrow. (**b**) Two abscess cavities are identified, one below the right hemidiaphragm, indicated with a red arrow, and one in the left paracolic gutter, indicated with a green arrow. The small bowel dilation is indicated by the blue arrow. (**c**) The cystic mass is indicated with a red arrow, with a maximum diameter of 6.74 cm.

**Figure 2 diagnostics-14-01975-f002:**
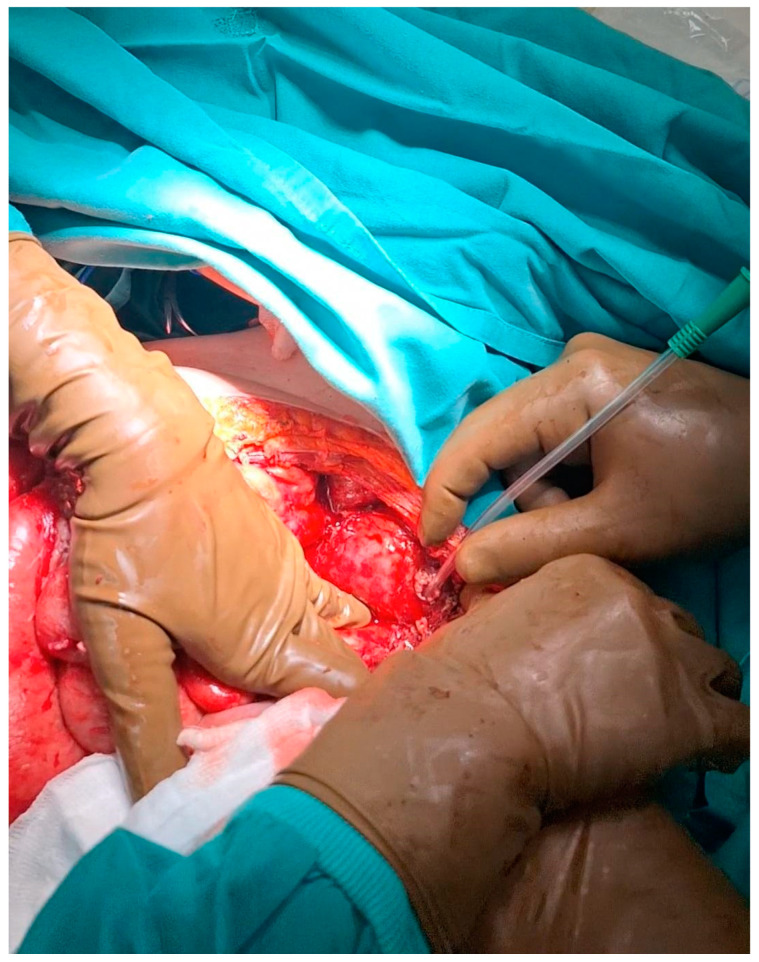
A cystic formation protruding from the right ovary was identified, with a rupture point on its wall, through which purulent fluid was leaking.

**Figure 3 diagnostics-14-01975-f003:**
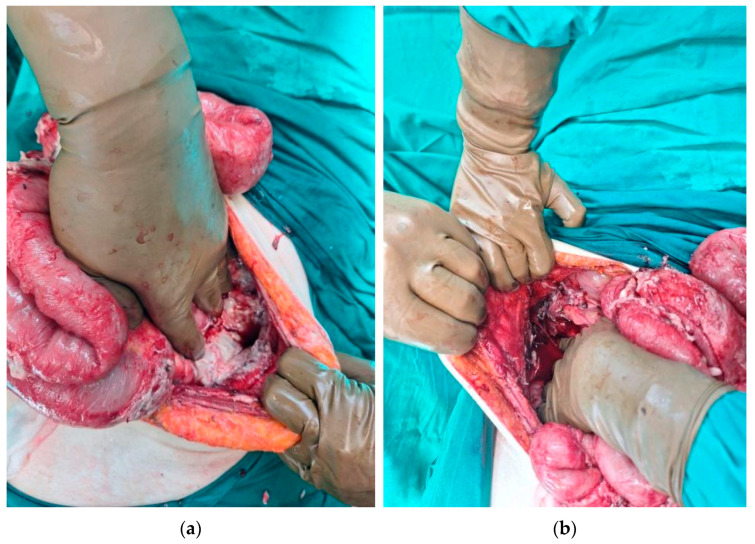
Intraoperative images showing the two abscess cavities. (**a**) An abscess cavity in the left paracolic area. (**b**) An abscess cavity below the right hemidiaphragm.

## Data Availability

All relevant data are within the manuscript.

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
