# Peer review of "Small Bowel Obstruction Masking a Perforated Dermoid Ovarian Cyst"

_diagnostics, 2024, doi:10.3390/diagnostics14171975_

Round 1

Reviewer 1 Report

Comments and Suggestions for Authors

Thank you for submitting your manuscript to Diagnostics! Here are my comments and indications:

Comment 1 Abstract section- “Imaging via CT scan revealed a large cystic mass in the right ovary, abscesses and generalized small bowel distension, which initially led to a diagnosis of ovarian cancer with peritoneal carcinomatosis.” This phrase does not sound good! You do not put a cancer diagnosis on CT, you raise a suspicion from the imaging data. The final diagnosis is histopathological. Please rephrase it.

Comment 2 Introduction- line 28 “elevated white blood cell count of 18.24x103/μL” please correct the WBC number as 18.24x103/μL.

Comment 3 Introduction- line 25 “wiht complains” please correct

Comment 4 – This case report has no structure. You should include the following sections: introduction, case presentation (subsections: clinical findings, imagistic findings, intraoperative findings, postsurgical evolution), discussions, conclusions.

Comment 5- The reference style is incorrect. Please use a citation software like Zotero or Endnote to include them, and adapt their format according to the MDPI template.

Comment 6- In the discussion section you should point out some differential diagnoses and point out the case particularities.

Comment 7- There are some extra spaces in the main text that should be removed.

Comment 8- The images’ resolution should be at least 300 dpi. Please revise.

Comments on the Quality of English Language

Minor English editing is required

Author Response

Comment 1 Abstract section- “Imaging via CT scan revealed a large cystic mass in the right ovary, abscesses and generalized small bowel distension, which initially led to a diagnosis of ovarian cancer with peritoneal carcinomatosis.” This phrase does not sound good! You do not put a cancer diagnosis on CT, you raise a suspicion from the imaging data. The final diagnosis is histopathological. Please rephrase it.

response 1 - we will revise

Comment 2 Introduction- line 28 “elevated white blood cell count of 18.24x103/μL” please correct the WBC number as 18.24x103/μL.

response 2 - we will revise

Comment 3 Introduction- line 25 “wiht complains” please correct

response 3 - we will revise

Comment 4 – This case report has no structure. You should include the following sections: introduction, case presentation (subsections: clinical findings, imagistic findings, intraoperative findings, postsurgical evolution), discussions, conclusions.

response 4 - this case report is submitted for special issue Interesting images. Please check the instructions for authors for this special issue, where you can see that no structure should accompany the images. The editor asked us to remove these sections in order for the manuscript to fit the submission guidelines for this issue.

Comment 5- The reference style is incorrect. Please use a citation software like Zotero or Endnote to include them, and adapt their format according to the MDPI template.

response 5 - thank you for your comment

Comment 6- In the discussion section you should point out some differential diagnoses and point out the case particularities.

response 6 - we will add the information

Comment 7- There are some extra spaces in the main text that should be removed.

response 7 - we will revise

Comment 8- The images’ resolution should be at least 300 dpi. Please revise.

response 8 - we will revise

Reviewer 2 Report

Comments and Suggestions for Authors

The authors present a clinical case of a woman who arrived to the emergency Department with perforated dermoid ovarian cyst which presented clinically as a small bowel obstruction due to peritoneal carcinomatosis of ovarian origin. 

The manuscript is well presented and with a nice iconography.

The clinical case illustrates the diverse clinical patterns of presentation of a ovarian dermoid cyst and emphasized that this entity may taken into consideration in the differential diagnosis of bowel obstrution suspecting peritoneal carcinomatosis.

The manuscript is interesting, but is neither new nor exceptional. I do not find sound arguments for the publication of this version of this manuscript.

I suggest that the authors to modify the Discussion section in order to include an extensive review of the clinical patterns of presentation of the ovarian cyst and its diagnostic options.

Comments on the Quality of English Language

English languaje is fine

Author Response

Comment 1 The authors present a clinical case of a woman who arrived to the emergency Department with perforated dermoid ovarian cyst which presented clinically as a small bowel obstruction due to peritoneal carcinomatosis of ovarian origin. 

The manuscript is well presented and with a nice iconography.

The clinical case illustrates the diverse clinical patterns of presentation of a ovarian dermoid cyst and emphasized that this entity may taken into consideration in the differential diagnosis of bowel obstrution suspecting peritoneal carcinomatosis.

The manuscript is interesting, but is neither new nor exceptional. I do not find sound arguments for the publication of this version of this manuscript.

I suggest that the authors to modify the Discussion section in order to include an extensive review of the clinical patterns of presentation of the ovarian cyst and its diagnostic options.

Response 1: Thank you for your comment. We will add a more extensive review of the clinical patterns of presentation of the ovarian cysts and its diagnostic options, as suggested.

Reviewer 3 Report

Comments and Suggestions for Authors

As recommendations for the authors would be the introduction of an abdominal radiograph, if there is a better marking with arrows on the CT images of intra-abdominal collections and ileal dilatation.

Author Response

As recommendations for the authors would be the introduction of an abdominal radiograph, if there is a better marking with arrows on the CT images of intra-abdominal collections and ileal dilatation.

Thank you for your comment.

We will add the required information and images.

Round 2

Reviewer 2 Report

Comments and Suggestions for Authors

The authorthe recomendations have modified the text of the manuscript following the suggested recommendations.

The manuscript is greatly improved. In my opinion it is suitable for publication.

Comments on the Quality of English Language

The english languaje is fine, only minor revision is needed